# Bridging the Operational Efficiency Differences between Franchisors and Franchisees: A Metafrontier Approach

Seung Beom Kim [1] and Kanghwa Choi [2],*

1 Department of Smart Convergence Consulting, Hansung University, Seoul 02876, Korea
2 Department of Business, Hansung University, Seoul 02876, Korea
* Correspondence: khchoi@hansung.ac.kr; Tel.: +82-02-760-8015

**Abstract:** A franchise business is a contractual relationship in which the franchisor and franchisee should cooperate to promote sustainable growth of their franchise entities. However, it is still unclear whether the relationship between franchisees and franchisors is a principal–agent or a business partner sharing a business goal. Thus, this study is a first attempt to investigate the relationship between franchisees and franchises using metafrontier and bootstrap DEA from the perspective of efficiency. We measured the efficiency of coffee franchises in Korea, which have grown rapidly in recent years despite the COVID-19 pandemic. Based on the bootstrap DEA results, there was a statistically significant difference in efficiency between franchisors and franchisees under the variable return-to-scale assumption. While the main cause of inefficiency in premium coffee chains is attributed to scale inefficiency, most franchisees have pure technological inefficiency. Thus, coffee franchisees can improve the operational efficiency by adjusting their business scale and reallocating service resources. This study demonstrates tailored operational plans to improve the operational efficiency of premium and mainstream coffee franchises and offers strategic initiatives to decrease the difference in efficiency between franchisors and franchisees.

**Keywords:** coffee franchise; premium and mainstream brand; metafrontier DEA; Mann–Whitney U test

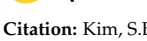

## 1. Introduction

The coffee franchise market in Korea is one of the fastest evolving and growing markets among various service franchise industries [1]. In particular, the coffee franchise industry in Korea is demonstrating a particularly strong upward trend due to the expansion of low-cost coffee franchise outlets [2], despite the recent COVID-19 pandemic. According to the statistical data released by the Korean National Tax Service (www.nts.go.kr (accessed on 10 September 2022)), the number of registered coffee and beverage outlets in 2021 was 77,543, an increase of approximately 16.6% compared to 2020. Mega MGC Coffee Franchise, a representative low-priced mainstream coffee shop, started to grow rapidly from 2020 and operated about 1643 coffee outlets in 2021, ranking second among Korean coffee franchises based on the number of outlets (www.mega-mgccoffee.com (accessed on 5 September 2022)). In particular, the popularity of low-price coffee franchises continues to grow, as the number of customers focused on the price-to-quality ratio and takeout-oriented demand has skyrocketed.

In general, in a franchise business model, franchisors provide trademarks, intellectual property rights, and business know-how to franchisees, and franchisees sell goods and services under the support or control of franchisors and pay a specific percentage of sales revenue as royalty fees to franchisors [3–6]. That is, a franchise is a contractual relationship in which the franchisor and the franchisee should cooperate to carry out their business [7,8]. The franchisors and franchisees need to make mutual efforts to promote sustainable growth of the franchise business. However, in reality, conflicts between franchisors and franchisees

are escalating due to asymmetric information and the imbalance of status in their transaction relationships [9].

As shown in the causal loop diagram in Figure 1, a franchisor strives to increase the number of franchisees. Moreover, franchisors receive significant initial fee payments from franchisees, which increases franchisors' revenue (R1 Loop). In addition, as franchisors' revenue increases, they expand their investment in franchisees [3]. This investment expansion generates a virtuous cycle that increases the revenue of franchisees and ultimately increases the franchise model's royalty revenue (R2 Loop). Thus, the core of the franchise business model is to create a virtuous cycle in which the profitability of franchisors and franchisees increases in the same direction [4–6]. Conversely, from the perspective of franchisees, as their number increases due to the efforts of franchisors, competition among inter- and intra-franchisees intensifies, and as a result, the profitability of franchisees deteriorates (B1/B2 Loop). Clearly, there is a strategic goal incongruity between franchisees and franchisors, which is the cause of conflict between them [3–7,10–12].

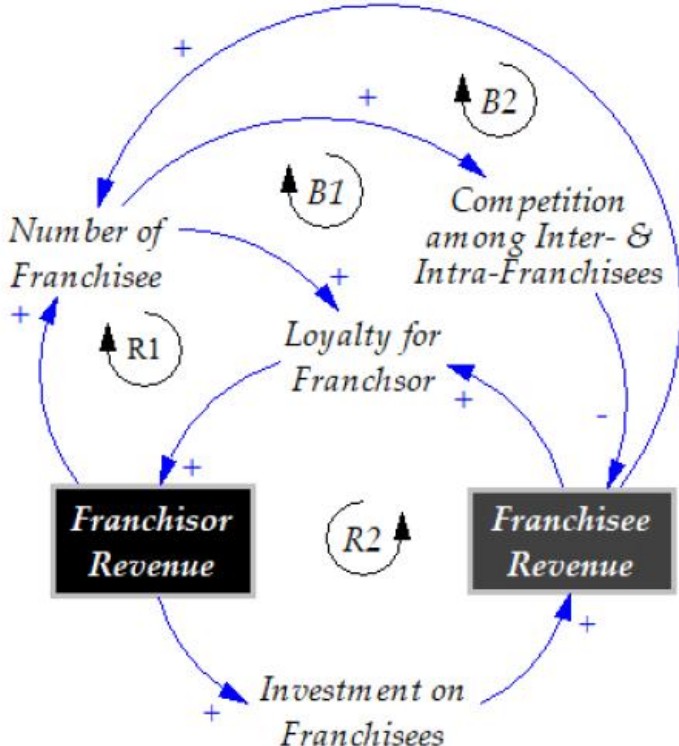

**Figure 1.** A causal loop diagram of the relationship between franchisors and franchisees.

Thus, this study aims to examine the relationship between franchisees and franchisors in terms of efficiency. That is, we intend to investigate the relationship of the virtuous cycle between the efficiency of franchisors and that of franchisees. Most previous studies related to franchise business have separately measured the efficiency of franchisees or franchisors [1–4,10–14]. In particular, the prior applications of data envelopment analysis (DEA) models on the relationship between franchisees and franchisors have been scarce. Thus, this study examined any difference between the efficiency of franchisors and franchisees using 28 coffee brand franchises in Korea.

Moreover, this study categorized Korean coffee shop franchises into two groups, *premium* and *mainstream* coffee shop franchises, based on their business scope (e.g., the number of franchisees) and specific operational strategies [15]. These two types of coffee franchises have different strategic outlet operation plans, specific customer responses, and differentiated price ranges for coffee and beverages. Therefore, this study used the metafrontier DEA to compare the relative efficiency between different groups with heterogeneous production frontiers [1,15,16].

The specific research issues of interest covered in this study are as follows:

(1) Research Question 1: What are the main causes of the observed inefficiency in coffee chains in Korea?

(2) Research Question 2: Is there a significant relationship in the operational efficiency between franchisors and franchisees? Does the operational efficiency between franchisors and franchisees have the same direction?

(3) Research Question 3: Is there any difference between the operating efficiency of premium and mainstream brand coffee chains in Korea?

This study makes the following theoretical and practical contributions:

First, it is the first attempt to link complicated and sophisticated metafrontier DEA to a principal and agent relationship in the coffee franchise agreement and compare the metafrontier values between franchisors and franchisees to distinguish their intricate inter-relationship in more detail. Second, this study addresses that premium and mainstream coffee franchises require differentiated initiatives tailored to their operational plans to maximize their efficiency. The premium coffee chains should be run in a store-centric high-end way, whereas mainstream chains should concentrate on low-price-oriented operating strategies. Third, franchisors and franchisees with heterogeneous production technologies have different causes of inefficiency. Thus, premium coffee chains should alter operating scales to minimize operational inefficiencies. In addition, coffee franchisees need to reallocate their operating resources to improve managerial efficiency.

The rest of this paper is organized as follows. Section 2 briefly summarizes previous studies on the efficiency of the coffee franchise business using the DEA approach. The research model and empirical data used in this study are introduced in Section 3. Section 4 summarizes the empirical results obtained from the application of metafrontier and bootstrap DEA. Section 5 discusses the theoretical and practical implications as well as the study's limitations.

## 2. Literature Review of Food Franchise Efficiency

In the franchise industries literature, several previous studies have assessed food franchise efficiency via non-parametric DEA in various food service fields. Several studies have measured the efficiency of coffee franchises [1,2,11–14] and food restaurant chains [3,16–18]. Nonetheless, until now, prior applications of DEA models on coffee franchises have been scarce. In particular, as most existing studies on coffee franchises focused on franchisees or franchisors independently, there was a limit to scrutinizing the interconnection of efficiency between franchisors and franchisees. As a result, to analyze the relationship between the efficiency of franchisors and franchisees, this study examined both franchisors and franchisees, two parties in a franchise business, as decision-making units (DMUs) for the metafrontier DEA. Table 1 summarizes existing studies on food franchise efficiency, their methodologies, the characteristics of DMUs, and associated input and output variables.

In this study, the empirical application deals with the coffee franchise sector in Korea and therefore, looked closely at the previous literature related to coffee franchises' efficiency. Park et al. [1] categorized 29 Korean coffee shop franchisors into three groups depending on their number of franchisees: small-, medium-, and large chains. In addition, they adopted the metafrontier DEA to measure and compare the efficiency of coffee shop franchisors. Joo et al. [12] assessed the retail operations of eight coffee shops owned by a specialty coffee company using input-oriented DEA and suggested that the location of a coffee shop is a crucial factor in assessing its profit efficiency. In addition, Kim et al. [11] measured the relative efficiency of six famous coffee franchises in Korea: *Starbucks*, *Coffee Bean*, *Cafe Bene*, *Ediya*, *Hollys*, and *Tom N Toms*. Recently, Wang et al. [14] employed network metafrontier DEA to estimate the cross-strait performances of the 54 B local coffee company in Taiwan and China from 2010 to 2012. In particular, they measured the difference in efficiencies between the outlet and business channel in Taiwan and China.

**Table 1.** Literature review of coffee franchise efficiency.

|  | Authors | Method | DMUs | Input Variables | Output Variables |
|---|---|---|---|---|---|
| Coffee Industry | Park et al. [1] | Metafrontier DEA | 29 Korean Coffee Shop Franchisors | Employee, Franchisee's Average Sales, Number of Franchisee | Financial Stabilization, Total Sales, Total Asset |
|  | Kim et al. [11] | Input-oriented CCR | 6 Coffee Franchises in Korea from 2010 to 2014 | Product cost, Labor cost, Costs for interior design | Sales Profit |
|  | Joo et al. [12] | Input-oriented CCR/BCC | 8 Premier Coffee Retail Stores | Cost of Sales, Occupancy expenses, Wages/Benefits, Other Expenses | Total Sales |
|  | Joo et al. [13] | Input-oriented CCR | 7 Specialty Coffee Retailers | Costs of Goods Sold, Sales, General, and Administrative Expenses, Depreciation/Amortization | Revenue |
|  | Wang et al. [14] | Network metafrontier DEA | 54 B Coffee Company of Taiwan and China from 2010 to 2012 | Personnel Pay, Raw Material Costs, Selling Expenses | Operating Income, Sales Revenue, Advance Receipts, Administrative Expenses |
| Food Restaurant Industry | Reynolds [17] | CCR/ BCC | 38 chains of same-brand franchises in United States | Hours worked, Average wage, Number of competitors, Seating capacity | Daily sales, Tip percentage |
|  | Roh and Choi [3] | Input-oriented CCR/ BCC | 550 chain restaurants operating within the Pacific Rim | Environmental/Location, Physical resources, Human resources, Management efficiency | Sales, Net income |
|  | Sveum and Sykuta [18] | Two-stage DEA | 8900 restaurants in United States | Payroll, Age of the establishment, Number of seats | Total Sales, Counter Sales, Drive-thru Sales, Takeout Sales, Server Sales |
|  | Alberca and Parte [19] | Metafrontier DEA | 863 restaurants in Spain | Total assets, Staff Costs, Cost of Sales | Total Sales |

In previous studies on the efficiency of food restaurants, Reynolds [17] analyzed 38 same-brand midscale restaurants located throughout the northeastern United States and suggested that DEA analysis has utility for food service operators to accurately assess productivity. Further, Roh and Choi [3] used the DEA methodology to empirically compare and contrast the efficiency of multiple brands within the same franchise in the Pacific Rim. Moreover, Alberca and Parte [19] employed metafrontier DEA to investigate operational efficiency affected by restaurant size in the restaurant business. Sveum and Sykuta [18] estimated the efficiency differences between franchisee-owned and franchisor-owned restaurants in full- and limited-service restaurants by using the two-stage DEA model.

The aforementioned previous studies related to coffee franchises are mainly focused on franchisees' or stores' efficiency [1,11–14]. Meanwhile, this study aims to clarify the relationship between the operational efficiency of an established brand (franchisor, parents company) and an independent business owner (franchisee) [5,6].

### 3. Research Model and Data

This study aims to highlight the relationship between the operational efficiency of franchisors and franchisees in Korean coffee chains and compare the operational efficiency of premium and mainstream coffee chain groups. For the metafrontier DEA, a total of 28 coffee chain brands (9 premium brands and 19 mainstream brands) were set as DMUs. In Korea, there are more coffee franchises than the DMUs of coffee chains employed in this study. However, this study considered all coffee franchises highly ranked in the Korean coffee chain as DMUs, except for some franchises that do not disclose financial and non-financial information (e.g., *Starbucks*, *PaulBasset*, and *Caffe-Pascucci*). See Appendix A for the DMU codes, full names of coffee chain groups, and their business histories.

The research model for metafrontier DEA is depicted in Figure 1. As seen in Figure 2, Korean coffee chains can be categorized into premium and mainstream coffee shop franchises, according to their operating alternatives, service differentiation, and served coffee price range. The premium coffee chains, focused on luxury and specialty coffee, serve coffee brewed by expert baristas using specialty grade beans, with a comparatively high price of more than USD 4 for Americano coffee [1,2]. Meanwhile, mainstream coffee shops offer limited services in casual and friendly settings at a low price of less than USD 2 a cup, and are primarily distinguished by consumers who seek price-to-quality through delivery and takeout services.

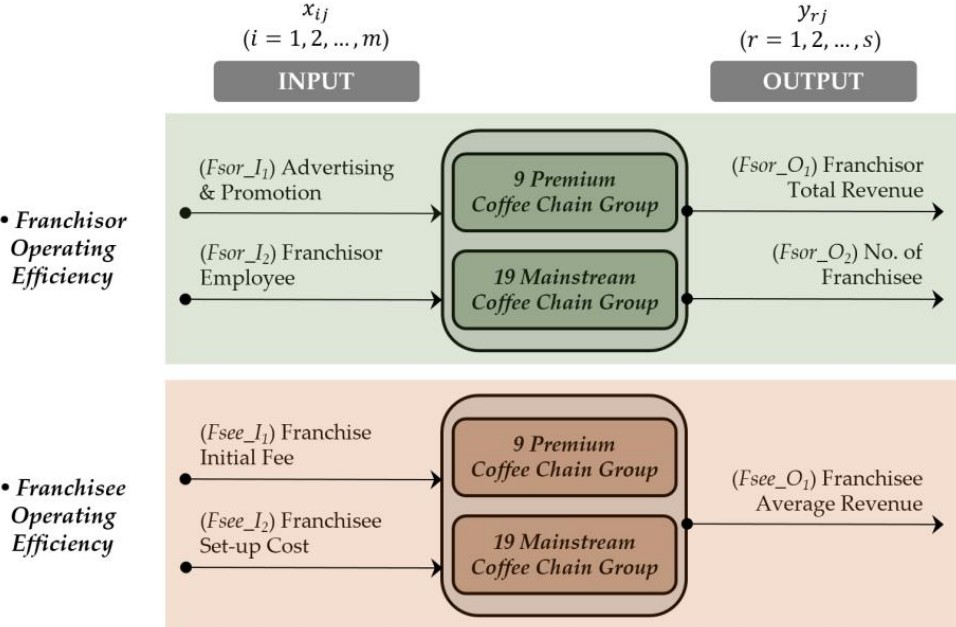

**Figure 2.** Research model for the metafrontier DEA on Koran coffee shop franchises.

To measure the meta-efficiency (ME) of 28 coffee brands' franchises in Korea, we obtained the financial and non-financial data for the year 2020 from the Korea Fair Trade Commission (franchise.ftc.go.kr (accessed on 5 September 2022)). Based on the previous literature review and data availability, this study divided input/output variables for measuring the efficiency of franchisors and franchisees. For measuring the ME of franchisors, this study employed (Fsor_$I_1$) franchisors' advertising and promotion expenses, (Fsor_$I_2$) franchisors' employee as input, and (Fsor_$O_1$) franchisors' total sales, (Fsor_$O_2$) number of franchisee as output variables. In addition, this study selected (Fsee_$I_1$) franchisees' initial fee, (Fsee_$I_2$) franchisees' set-up cost as input, and (Fsee_$O_1$) franchisees' average sales as output variables to measure the ME of the franchisee [1,12–15].

The definitions of input/output variables are as follows:

- Franchisor's Advertising and Promotion Expenses (FsorAP): The costs for promoting a franchise brand and recruiting franchisees.

- Franchisor Employee (FsorE): Full-time employees of a parent franchising company, such as employees in any other type of business (e.g., franchise supervisors and franchise operating managers).
- Franchisor Total Sales (FsorTS): The annual total sales amount of a franchise establishment company, including: (a) franchisees' sales-based ongoing royalties, (b) franchise initial fee revenue, and (c) distribution revenue allocated to goods and services that franchisees sell.
- Number of Franchisees (NFsee): The total number of franchised outlets under the control of a parent franchising company.
- Franchise Initial Fee (FseeIF): The license fees of entry and owning a franchise to be paid by a franchisee to a franchising company (e.g., franchise subscription fee, education fee, deposit, etc.).
- Franchisee Set-up Cost (FseeSC): This refers to the total cost of opening a franchise shop, specifically, the interior costs of a franchisee, such as the cost of supplies and equipment, furniture, and fixtures.
- Franchisee Average Sales (FseeAS): The average annual sales per franchisee over a given time.

Most franchisors provide not just trade name, products, and services but also an entire system for operating the business to their franchisees (www.franchise.org (accessed on 5 September 2022)). Franchisors strive to boost the number of franchisees and encourage sustainable growth of the franchise business by maximizing its limited resources. Franchisees pay the franchise initial fee and set-up cost to their franchisor for the right to conduct the franchised business. Then, franchisees attempt to maximize their stores' revenue through franchisors' ongoing administrative or technical support (e.g., human resources and accounting) [3–7,11–14]. Thus, this study adopted the output-oriented DEA model to empirically measure contemporaneous efficiency of Korean coffee brand franchises. With the output-oriented DEA, the linear program is configured to maximize a firm's potential output without requiring more of any observed input values [20].

Table 2 tabulates the descriptive statistics related to input/output variables adopted in the metafrontier DEA model. Comparing premium with mainstream coffee chains, premium brands have overwhelmingly higher advertising and promotion costs and more employees than mainstream brands. Moreover, the initial and set-up cost of premium coffee franchises is greater than that of mainstream franchises.

**Table 2.** Input and output variables in the Korean coffee shop franchises.

| | Category | Operational Efficiency for Franchisor | | | | Operational Efficiency for Franchisee | | |
| --- | --- | --- | --- | --- | --- | --- | --- | --- |
| | | FsorAP (1000 won) | FsorE (Person) | FsorTS (1000 won) | NFsee | FseeIF (1000 won) | FseeSC (1000 won) | FseeAS (1000 won) |
| Max | Premium | 13,342,110 | 2303 | 364,058,442 | 2885 | 39,150 | 260,050 | 508,907 |
| | Mainstream | 1,855,012 | 281 | 134,708,794 | 1188 | 21,000 | 156,051 | 298,566 |
| Min | Premium | 13,142 | 14 | 3,945,604 | 99 | 5300 | 51,590 | 90,685 |
| | Mainstream | 6743 | 11 | 1,047,124 | 105 | 0 | 42,400 | 34,789 |
| Ave | Premium | 3,018,980 | 456 | 93,293,856 | 668 | 18,561 | 142,369 | 220,237 |
| | Mainstream | 376,379 | 49 | 21,502,480 | 375 | 10,961 | 66,203 | 147,957 |
| SD | Premium | 4,584,804 | 732 | 126,456,525 | 916 | 9323 | 72,501 | 126,016 |
| | Mainstream | 515,076 | 62 | 31,121,487 | 300 | 5144 | 25,326 | 71,986 |

To assess the strength of the isotonic relationship between input and output variables, this study adopted Pearson correlation coefficients, as shown in Figure 3. All Pearson correlation coefficients are estimated to be positive, indicating that under the same condition, the output cannot decrease if the input increases. All DMUs meet the requirement that the number of DMUs should be greater than or equal to double the number of inputs plus outputs [20,21]. In this study, we used the MaxDEA Ultra (8.2 ver.) software to measure the metafrontier DEA estimators (maxdea.com (accessed on 20 August 2022)).

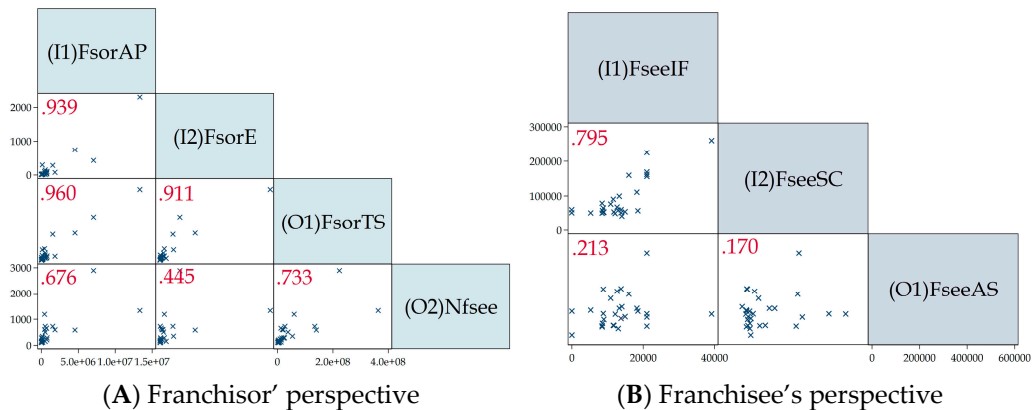

**Figure 3.** Pearson correlation analysis between input and output variables: (**A**) franchisor's perspective; (**B**) franchisee's perspective.

## 4. Results

The efficiency of DEA often refers to relative efficiency; efficiency score of individual DMUs is measured by a function of distance from the production frontier. Therefore, the efficiency change in individual DMUs is attributable to the following two attributes. First, the technology level of individual DMUs has changed due to technological innovation, and such fluctuations have resulted in upward and downward movement of the production frontier, which ultimately led to changing the efficiency of individual DMUs. Generally, this efficiency change is caused by arbitrary innovations of individual DMUs. Second, even if individual DMUs did not make any effort to change their efficiency, the production frontier has shifted upward and downward due to the change in efficiency of other DMUs within the group, and the relative efficiency of individual DMUs has changed accordingly. As a result, from the perspective of individual DMUs, it is possible to prepare a strategic operation plan for specific DMUs only when it is accurately ascertained whether the efficiency change is caused by an intentional or accidental fluctuation. The efficiency change related to the metafrontier DEA can be decomposed into two components: the change of distance from an input–output point to the group frontier or the change of distance between the group frontier and the metafrontier [1–3,21–24]. As mentioned by Piot-Lepetit [4] and Perrigot et al. [25], the results of metafrontier DEA provide useful information for benchmarking purposes by measuring ME, GE, and TGR simultaneously. Thus, this study measured ME, group efficiency (GE), and technology gap ratio (TGR) using the metafrontier DEA as a methodology. A detailed description of the metafrontier DEA methodology adopted in this study is presented in Appendix B.

### 4.1. Metafrontier DEA Results of Coffee Franchisors

Park et al. [1] classified Korean coffee shop franchisors into three groups according to their number of franchisees: large- (n ≥ 300), medium- (100 ≤ n < 300), and small coffee shop chains (n < 300). However, it is more meaningful to divide the coffee chains into premium and mainstream brands by reflecting on the operational features rather than dividing the groups according to the number of outlets. Thus, this study measured the ME of 28 homogeneous coffee shop franchises in Korea, then categorized coffee brand franchises into two heterogeneous groups depending on their operational characteristics and price ranges to assess their GE, and finally calculated the TGR of all DMUs.

The results of the metafrontier DEA under the constant return-to-scale (CRS) and variable return-to-scale (VRS) assumptions are given in Table 3 below. First, regardless of the return-to-scale assumption, the mainstream coffee brand franchisors have a higher average ME score (CRS-based average TE = 0.6423, VRS-based average PTE = 0.7468) than premium brand franchisors (TE = 0.3401, PTE = 0.7100), indicating that the mainstream coffee shop brands in Korea were more efficiently operated than premium coffee shop brands. Meanwhile, the TGR of mainstream coffee franchisors is closer to 1 (CRS-based:

1.00, VRS-based: 0.9998), demonstrating that there is little technological gap between ME and GE. Conversely, as the TGR value of premium coffee franchisors is relatively low (CRS-based average TGR = 0.4802, VRS-based average PTE = 0.7945), there is a significant difference between ME and GE. In particular, efficient DMUs, such as A(03), A(04), and A(06) with GE values of 1, showed relatively low ME values, mainly due to the greater technological gap between group frontier and metafrontier. These results demonstrate that DMUs in premium brands were highly efficient within the premium group, but the technological gap widened as they moved away from the production frontier that enveloped all DMUs.

**Table 3.** Meta-efficiency of franchisors.

| Group | DMU Code | CRS | | | VRS | | | SE | RTS | Main Cause of Inefficiency | |
|---|---|---|---|---|---|---|---|---|---|---|---|
| | | ME (TE) | GF | TGR | ME (PTE) | GF | TGR | | | PTE | SE |
| 1 | Fsor_A(01) | 0.0932 | 0.2130 | 0.4377 | 0.1602 | 0.4200 | 0.3814 | 0.5821 | DRS | ○ | |
| | Fsor_A(02) | 0.2168 | 0.5288 | 0.4099 | 0.3133 | 0.5638 | 0.5556 | 0.6919 | DRS | ○ | |
| | Fsor_A(03) | 0.4564 | 1 | 0.4564 | 1 | 1 | 1 | 0.4564 | DRS | | ○ |
| | Fsor_A(04) | 0.5730 | 1 | 0.5730 | 1 | 1 | 1 | 0.5730 | IRS | | ○ |
| | Fsor_A(05) | 0.2176 | 0.5174 | 0.4205 | 0.3527 | 0.6622 | 0.5327 | 0.6169 | DRS | ○ | |
| | Fsor_A(06) | 0.6448 | 1 | 0.6448 | 0.8319 | 1 | 0.8319 | 0.7751 | IRS | | ○ |
| | Fsor_A(07) | 0.4903 | 0.8849 | 0.5541 | 1 | 1 | 1 | 0.4903 | DRS | | ○ |
| | Fsor_A(08) | 0.1685 | 0.4092 | 0.4117 | 1 | 1 | 1 | 0.1685 | DRS | | ○ |
| | Fsor_A(09) | 0.2004 | 0.4846 | 0.4136 | 0.7321 | 0.8628 | 0.8485 | 0.2737 | DRS | | ○ |
| | *Ave.* | *0.3401* | *0.6709* | *0.4802* | *0.7100* | *0.8343* | *0.7945* | *0.5142* | *DRS: 77.8%, IRS: 22.2%* | | |
| 2 | Fsor_B(01) | 1 | 1 | 1 | 1 | 1 | 1 | 1 | CRS | | |
| | Fsor_B(02) | 0.4658 | 0.4658 | 1 | 0.7006 | 0.7026 | 0.9972 | 0.6649 | DRS | | ○ |
| | Fsor_B(03) | 0.3283 | 0.3283 | 1 | 0.3494 | 0.3494 | 1 | 0.9397 | IRS | ○ | |
| | Fsor_B(04) | 0.3319 | 0.3319 | 1 | 0.3353 | 0.3353 | 1 | 0.9897 | DRS | ○ | |
| | Fsor_B(05) | 0.7803 | 0.7803 | 1 | 1 | 1 | 1 | 0.7803 | DRS | | ○ |
| | Fsor_B(06) | 1 | 1 | 1 | 1 | 1 | 1 | 1 | CRS | | |
| | Fsor_B(07) | 1 | 1 | 1 | 1 | 1 | 1 | 1 | CRS | | |
| | Fsor_B(08) | 0.5137 | 0.5137 | 1 | 1 | 1 | 1 | 0.5137 | DRS | | ○ |
| | Fsor_B(09) | 0.3999 | 0.3999 | 1 | 0.4429 | 0.4429 | 1 | 0.9029 | DRS | ○ | |
| | Fsor_B(10) | 1 | 1 | 1 | 1 | 1 | 1 | 1 | CRS | | |
| | Fsor_B(11) | 0.6135 | 0.6135 | 1 | 0.6356 | 0.6362 | 0.9991 | 0.9651 | IRS | ○ | |
| | Fsor_B(12) | 0.5928 | 0.5928 | 1 | 0.7138 | 0.7138 | 1 | 0.8304 | DRS | ○ | |
| | Fsor_B(13) | 0.8584 | 0.8584 | 1 | 1 | 1 | 1 | 0.8584 | IRS | | ○ |
| | Fsor_B(14) | 0.4324 | 0.4324 | 1 | 0.4702 | 0.4702 | 1 | 0.9197 | DRS | ○ | |
| | Fsor_B(15) | 0.2301 | 0.2301 | 1 | 0.5767 | 0.5767 | 1 | 0.3991 | DRS | | ○ |
| | Fsor_B(16) | 0.5963 | 0.5963 | 1 | 0.7972 | 0.7972 | 1 | 0.7480 | DRS | | ○ |
| | Fsor_B(17) | 0.5614 | 0.5614 | 1 | 0.6314 | 0.6314 | 1 | 0.8891 | IRS | ○ | |
| | Fsor_B(18) | 1 | 1 | 1 | 1 | 1 | 1 | 1 | CRS | | |
| | Fsor_B(19) | 0.4995 | 0.4995 | 1 | 0.5364 | 0.5364 | 1 | 0.9312 | DRS | ○ | |
| | *Ave.* | *0.6423* | *0.6423* | *1* | *0.7468* | *0.7470* | *0.9998* | *0.8586* | *DRS: 71.4%, IRS: 28.6%* | | |

Second, the main causes of inefficiency of each DMU through comparison of PTE and SE values are as follows. In premium coffee franchisors, scale inefficiency (PTE > SE, 66.7%) is higher than pure technical inefficiency (PTE < SE, 33.3%), whereas in mainstream brands, pure technical inefficiency (57.1%) is higher than scale inefficiency (42.9%). This result demonstrates that premium franchisors require strategic alternatives to boost their efficiency by adjusting the scale of the economy. In particular, DMUs located in the DRS region (77.8%) make efforts to increase the operational performance as output factors, while DMUs in the IRS area should increase their efficiency by further expanding input factors. In addition, most franchises with pure technical inefficiencies have difficulties in the franchise operation system, such as allocating service operating resources inefficiently or underutilizing their management resources (i.e., *managerial inefficiency*). This is mainly

the case when franchisors' profitability or the number of franchisees does not increase compared to franchisors' investment in advertising expenses and promotion activities and the number of franchisors' executives and staff. Therefore, it is necessary to control the advertising and promotion of the budget through selection and concentration and to adjust the number of franchisors' employees. In particular, as the coffee franchise industry in Korea is already closer to market saturation, it is necessary to increase the number of franchisees by developing a new franchise business model [1,2].

### 4.2. Metafrontier DEA Results of Coffee Franchisees

From the perspective of the ME results of franchisees, the average ME value of premium franchise brands ($TE_{premium}$ = 0.4030) was lower than that of mainstream brands ($TE_{mainstream}$ = 0.4668) under the CRS assumption, as seen in Figure 4. Meanwhile, the average PTE score of premium franchisees ($PTE_{premium}$ = 0.5557) was higher than that of mainstream franchisee brands ($PTE_{mainstream}$ = 0.5178) under the VRS assumption. Looking at the critical causes of inefficient franchises, premium and mainstream franchisees had about 75.0% and 88.2% of pure technical inefficiencies, respectively. In particular, as inefficient DMUs were mostly located in the DRS area, it is necessary to improve their managerial alternatives to reduce operational inefficiency. The coffee franchise business in Korea has already reached a mature stage; therefore, it offers a relatively low return-on-investment. Thus, franchisees should attract customers in a different way from existing marketing policies and improve efficiency through new operating methods, such as delivery-oriented or self-service store operations [11–14].

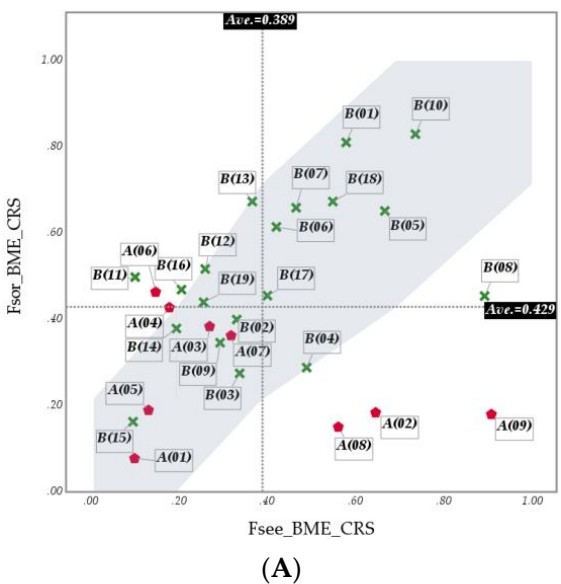

**(A)**

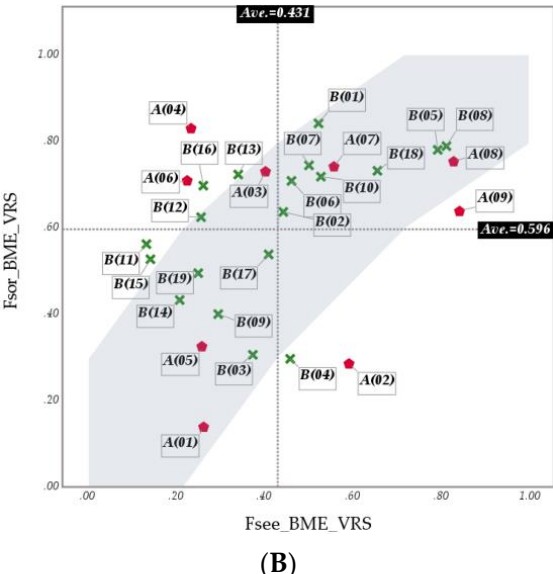

**(B)**

**Figure 4.** BME difference between franchisors and franchisees. ⬟: Premium Group, ✕: Mainstream Group; (**A**) CRS-based assumption; (**B**) VRS-based assumption.

### 4.3. Bootstrap DEA Results of Coffee Franchisees

In this study, we additionally attempt to analyze whether there is a difference in efficiency between franchisees and franchisors or between premium and mainstream coffee brands by using bootstrap DEA, as shown in Table 4 below. The bootstrap technique, a non-parametric statistical method, was first introduced by Efron [26] and has been widely used as an alternative to overcome the limitation of sample size and impracticality of parametric statistical methodology.

Table 4. Meta-efficiency of franchisees.

| Group | DMU Code | CRS | | | VRS | | | SE | RTS | Main Cause of Inefficiency | |
|---|---|---|---|---|---|---|---|---|---|---|---|
| | | ME (TE) | GF | TGR | ME (PTE) | GF | TGR | | | PTE | SE |
| 1 | Fsee_A(01) | 0.1104 | 0.1445 | 0.7642 | 0.3043 | 0.3043 | 1 | 0.3628 | DRS | ○ | |
| | Fsee_A(02) | 0.7280 | 1 | 0.7280 | 0.7787 | 1 | 0.7787 | 0.9350 | IRS | ○ | |
| | Fsee_A(03) | 0.3020 | 0.3907 | 0.7728 | 0.4539 | 0.4539 | 1 | 0.6652 | DRS | ○ | |
| | Fsee_A(04) | 0.1963 | 0.2625 | 0.7477 | 0.2584 | 0.2964 | 0.8718 | 0.7597 | DRS | ○ | |
| | Fsee_A(05) | 0.1508 | 0.2216 | 0.6805 | 0.3060 | 0.3060 | 1 | 0.4928 | DRS | ○ | |
| | Fsee_A(06) | 0.1626 | 0.2181 | 0.7454 | 0.2713 | 0.2713 | 1 | 0.5991 | DRS | ○ | |
| | Fsee_A(07) | 0.3603 | 0.5032 | 0.7160 | 0.6287 | 0.6680 | 0.9412 | 0.5731 | DRS | | ○ |
| | Fsee_A(08) | 0.6165 | 0.8224 | 0.7497 | 1 | 1 | 1 | 0.6165 | DRS | | ○ |
| | Fsee_A(09) | 1 | 1 | 1 | 1 | 1 | 1 | 1 | | CRS | |
| | *Ave.* | *0.4030* | *0.5070* | *0.7672* | *0.5557* | *0.5889* | *0.9546* | *0.6671* | | *DRS: 87.5%, IRS: 12.5%* | |
| 2 | Fsee_B(01) | 1 | 1 | 1 | 1 | 1 | 1 | 1 | | CRS | |
| | Fsee_B(02) | 0.3636 | 0.3636 | 1.0000 | 0.4944 | 0.6262 | 0.7894 | 0.7354 | DRS | ○ | |
| | Fsee_B(03) | 0.3712 | 0.3777 | 0.9830 | 0.4082 | 0.4332 | 0.9421 | 0.9096 | DRS | ○ | |
| | Fsee_B(04) | 0.5381 | 0.5501 | 0.9782 | 0.5648 | 0.5755 | 0.9814 | 0.9527 | IRS | ○ | |
| | Fsee_B(05) | 0.7399 | 0.7441 | 0.9943 | 0.8668 | 0.9580 | 0.9049 | 0.8535 | DRS | | ○ |
| | Fsee_B(06) | 0.4621 | 0.4621 | 1.0000 | 0.5177 | 0.5245 | 0.9871 | 0.8925 | DRS | ○ | |
| | Fsee_B(07) | 0.5107 | 0.5253 | 0.9723 | 0.5478 | 0.5711 | 0.9592 | 0.9324 | DRS | ○ | |
| | Fsee_B(08) | 1 | 1 | 1 | 1 | 1 | 1 | 1 | | CRS | |
| | Fsee_B(09) | 0.3236 | 0.3236 | 1.0000 | 0.3416 | 0.3437 | 0.9941 | 0.9473 | DRS | ○ | |
| | Fsee_B(10) | 0.8067 | 0.8296 | 0.9723 | 1 | 1 | 1 | 0.8067 | IRS | | ○ |
| | Fsee_B(11) | 0.1749 | 0.1749 | 1 | 0.2032 | 0.2032 | 1 | 0.8605 | DRS | ○ | |
| | Fsee_B(12) | 0.2859 | 0.2900 | 0.9858 | 0.2930 | 0.2979 | 0.9837 | 0.9757 | DRS | ○ | |
| | Fsee_B(13) | 0.4099 | 0.4101 | 0.9994 | 0.4126 | 0.4142 | 0.9962 | 0.9933 | DRS | ○ | |
| | Fsee_B(14) | 0.2140 | 0.2173 | 0.9845 | 0.2300 | 0.2406 | 0.9559 | 0.9302 | DRS | ○ | |
| | Fsee_B(15) | 0.1057 | 0.1057 | 1 | 0.1712 | 0.2856 | 0.5996 | 0.6173 | DRS | ○ | |
| | Fsee_B(16) | 0.2312 | 0.2312 | 1 | 0.2904 | 0.2986 | 0.9727 | 0.7960 | DRS | ○ | |
| | Fsee_B(17) | 0.4408 | 0.4534 | 0.9723 | 0.4573 | 0.4683 | 0.9765 | 0.9639 | DRS | ○ | |
| | Fsee_B(18) | 0.6038 | 0.6038 | 1 | 0.7281 | 0.8202 | 0.8877 | 0.8293 | DRS | ○ | |
| | Fsee_B(19) | 0.2868 | 0.2868 | 0.9999 | 0.3112 | 0.3112 | 1 | 0.9213 | IRS | ○ | |
| | *Ave.* | *0.4668* | *0.4710* | *0.9917* | *0.5178* | *0.5459* | *0.9437* | *0.8904* | | *DRS: 82.4%, IRS: 17.6%* | |

The conventional DEA model helps evaluate the efficiency by the linear distance function of each DMU without statistical assumptions; however, it also has the disadvantage of being a relative efficiency, in that the efficiency score changes whenever the number of DMUs fluctuates. That is, the traditional DEA does not perform statistical verification on the efficiency score, so the efficiency score may have a bias and may produce a distorted DEA score, in that it cannot offer a statistical confidence interval for the efficiency score. To overcome the difficulties of the non-parametric DEA method, Simar and Wilson [27,28] theoretically proposed a new parametric methodology for calculating the confidence interval and standard error by applying the bootstrap method to the DEA model. Accordingly, bootstrap DEA may explain the difference in efficiency between efficient DMUs even when there are multiple efficient DMUs.

Under the assumption of CRS and VRS, the bootstrap DEA results of the coffee franchisors and franchisees are shown in Table 5 below. According to the results of the bootstrap DEA under the CRS assumption, the DMUs of the premium brands' group showing a significant ME difference between franchisors and franchisees are A(02), A(06), A(08), and A(09), and the DMUs of mainstream brand groups are B(08), B(11) and B(13). In addition, under the VRS assumption, A(02), A(03), A(04), and A(06) in the premium group have a great difference in efficiency between franchisors and franchisees, and in the mainstream brand group, B(01), B(11), B(12), B(13), B(15), and B(16) also have a meaningful difference. In general, the efficiency between franchisors and franchisees should have a positive linear combination; however, these DMUs show a large discrepancy in efficiency.

This result is in line with the findings of previous studies by Garg et al. [7] and Perrigot et al. [29] in that both franchisors and franchisees with different goals and priorities face principal and agency problems. Consequently, these DMUs should strive to reduce this difference by adjusting the profit-sharing structure between franchisors and franchisees or more efficient control of franchisees [30].

**Table 5.** Bootstrap meta efficiency analysis (based on CRS and VRS) and differences in bootstrap meta efficiency (BME) between franchisor and franchisee.

| DMU Code | CRS-Based | | | | | VRS-Based | | | | |
|---|---|---|---|---|---|---|---|---|---|---|
| | Franchisor | | Franchisee | | Diff. | Franchisor | | Franchisee | | Diff. |
| | ME | BME (A) | ME | BME (B) | (A)-(B) | ME | BME (A) | ME | BME (B) | (A)-(B) |
| A(01) | 0.0932 | 0.0775 | 0.1104 | 0.0998 | (0.0223) | 0.1602 | 0.1383 | 0.3043 | 0.2638 | (0.1254) |
| A(02) | 0.2168 | 0.1837 | 0.7280 | 0.6459 | *(0.4622)* | 0.3133 | 0.2848 | 0.7787 | 0.5923 | *(0.3075)* |
| A(03) | 0.4564 | 0.3833 | 0.3020 | 0.2698 | 0.1135 | 1 | 0.7293 | 0.4539 | 0.4036 | *0.3256* |
| A(04) | 0.5730 | 0.4272 | 0.1963 | 0.1785 | 0.2487 | 1 | 0.8292 | 0.2584 | 0.2353 | *0.5939* |
| A(05) | 0.2176 | 0.1893 | 0.1508 | 0.1313 | 0.0580 | 0.3527 | 0.3247 | 0.3060 | 0.2598 | 0.0649 |
| A(06) | 0.6448 | 0.4631 | 0.1626 | 0.1475 | *0.3156* | 0.8319 | 0.7084 | 0.2713 | 0.2266 | *0.4818* |
| A(07) | 0.4903 | 0.3621 | 0.3603 | 0.3184 | 0.0437 | 1 | 0.7408 | 0.6287 | 0.5583 | 0.1825 |
| A(08) | 0.1685 | 0.1503 | 0.6165 | 0.5612 | *(0.4109)* | 1 | 0.7526 | 1 | 0.8289 | (0.0762) |
| A(09) | 0.2004 | 0.1797 | 1 | 0.9080 | *(0.7284)* | 0.7321 | 0.6376 | 1 | 0.8422 | (0.2047) |
| B(01) | 1 | 0.8100 | 1 | 0.5790 | 0.2311 | 1 | 0.8412 | 1 | 0.5235 | *0.3177* |
| B(02) | 0.4658 | 0.3995 | 0.3636 | 0.3312 | 0.0683 | 0.7006 | 0.6364 | 0.4944 | 0.4438 | 0.1926 |
| B(03) | 0.3283 | 0.2748 | 0.3712 | 0.3378 | (0.0630) | 0.3494 | 0.3058 | 0.4082 | 0.3750 | (0.0692) |
| B(04) | 0.3319 | 0.2882 | 0.5381 | 0.4896 | (0.2014) | 0.3353 | 0.2965 | 0.5648 | 0.4594 | (0.1629) |
| B(05) | 0.7803 | 0.6512 | 0.7399 | 0.6667 | (0.0155) | 1 | 0.7800 | 0.8668 | 0.7929 | (0.0129) |
| B(06) | 1 | 0.6142 | 0.4621 | 0.4210 | 0.1933 | 1 | 0.7081 | 0.5177 | 0.4624 | 0.2457 |
| B(07) | 1 | 0.6585 | 0.5107 | 0.4657 | 0.1928 | 1 | 0.7442 | 0.5478 | 0.5020 | 0.2422 |
| B(08) | 0.5137 | 0.4539 | 1 | 0.8927 | *(0.4388)* | 1 | 0.7887 | 1.0000 | 0.8126 | (0.0239) |
| B(09) | 0.3999 | 0.3459 | 0.3236 | 0.2935 | 0.0523 | 0.4429 | 0.4000 | 0.3416 | 0.2968 | 0.1032 |
| B(10) | 1 | 0.8291 | 0.8067 | 0.7359 | 0.0932 | 1 | 0.7179 | 1 | 0.5289 | 0.1891 |
| B(11) | 0.6135 | 0.4979 | 0.1749 | 0.1012 | *0.3966* | 0.6356 | 0.5618 | 0.2032 | 0.1344 | *0.4274* |
| B(12) | 0.5928 | 0.5167 | 0.2859 | 0.2598 | 0.2570 | 0.7138 | 0.6243 | 0.2930 | 0.2581 | *0.3662* |
| B(13) | 0.8584 | 0.6730 | 0.4099 | 0.3664 | *0.3066* | 1 | 0.7229 | 0.4126 | 0.3420 | *0.3808* |
| B(14) | 0.4324 | 0.3788 | 0.2140 | 0.1946 | 0.1842 | 0.4702 | 0.4325 | 0.2300 | 0.2096 | 0.2229 |
| B(15) | 0.2301 | 0.1627 | 0.1057 | 0.0963 | 0.0664 | 0.5767 | 0.5269 | 0.1712 | 0.1435 | *0.3835* |
| B(16) | 0.5963 | 0.4688 | 0.2312 | 0.2062 | 0.2625 | 0.7972 | 0.6971 | 0.2904 | 0.2631 | *0.4340* |
| B(17) | 0.5614 | 0.4549 | 0.4408 | 0.4006 | 0.0543 | 0.6314 | 0.5381 | 0.4573 | 0.4107 | 0.1274 |
| B(18) | 1 | 0.6729 | 0.6038 | 0.5489 | 0.1240 | 1 | 0.7315 | 0.7281 | 0.6566 | 0.0748 |
| B(19) | 0.4995 | 0.4395 | 0.2868 | 0.2561 | 0.1834 | 0.5364 | 0.4947 | 0.3112 | 0.2516 | 0.2431 |

### 4.3.1. Comparison between Franchisors and Franchisees

In this study, we additionally conducted the Mann–Whitney U test to analyze whether there were statistically significant differences in the bootstrap meta-efficiency (BME) scores of franchisors and franchisees. The test results are presented in Table 6 and Figure 4.

The results demonstrated that, under the CRS assumption, there was no significant difference in BME between franchisors and franchisees (Asymp. Sig = 0.342 > 0.05), as seen in Table 6A. Conversely, in the VRS-based BME difference analysis in Table 6B, there was a statistically significant difference between franchisors and franchisees at the 5% significance level (Mann–Whitney U = 221.0, Wilcoxon W = 627.0, Asymptotic Sig. = 0.005 < 0.05). The mean rank of coffee franchisors (mean rank = 34.61) was higher than that of coffee franchisees (mean rank = 22.39), indicating that the average efficiency of franchisors was overall higher than that of franchisees, such as the results of the VRS-based ME analysis.

**Table 6.** Analysis result of Mann–Whitney difference between franchisors and franchisees.

| (A) CRS-Based | (B) VRS-Based |
|---|---|

Mann–Whitney U = 334.0,
Wilcoxon W = 740.0, S.E = 61.025,
Asymptotic Sig. = 0.342

Mann–Whitney U = 221.0,
Wilcoxon W = 627.0, S.E = 61.025,
Asymptotic Sig. = 0.005 ***

*** denotes 1% significant level.

### 4.3.2. Comparison between Premium and Mainstream Brand Groups

We performed the Mann–Whitney U test to compare the BME scores of premium and mainstream franchise groups, as seen in Table 7. The result showed that, under the CRS assumption, there was a statistically significant difference in efficiency between premium and mainstream coffee brand franchisors at the 5% significance level (Mann–Whitney U = 144.0, Wilcoxon W = 334.0, Asymp. Sig = 0.004 < 0.05). Moreover, the mean rank of premium brand franchisors (mean rank = 8.00) was lower than that of mainstream brand franchisors (mean rank = 17.58). Meanwhile, there were no statistically significant differences between premium and mainstream brand franchises in the VRS-DEA.

**Table 7.** Analysis result of Mann–Whitney U test between premium and mainstream coffee chain.

| | CRS-Based | | VRS-Based | |
|---|---|---|---|---|
| | **Franchisor** | **Franchisee** | **Franchisor** | **Franchisee** |
| Total *n* | *n = 28 (Premium = 9, Mainstream = 19)* | | | |
| Mean Rank * | *P = 8.00, M = 17.58* | *P = 12.78, M = 15.32* | *P = 14.67, M = 14.42* | *P = 15.67, M = 13.95* |
| Mann–Whitney U | *144.000* | *101.000* | *84.000* | *75.000* |
| Wilcoxon W | *334.000* | *291.000* | *274.000* | *265.000* |
| Standardized Test | *2.878* | *0.762* | *−0.074* | *−0.517* |
| Asymptotic Sig. | *0.004 **** | *0.446* | *0.941* | *0.605* |

* Note: *P* indicates premium group, and *M* indicates mainstream. *** denotes 1% significant level.

## 5. Discussion

### 5.1. Theoretical Implications

This study contributes to the literature concerning the efficiency of the coffee franchise industry. Despite the economic recession caused by the COVID-19 pandemic, coffee franchise brands in Korea are still growing, as mainstream coffee chains have skyrocketed in number. The market size of the coffee franchise industry in Korea rapidly increased from USD 300 million in 2007 to USD 4.3 billion in 2018, ranking third in the world after the United States (USD 26.1 billion) and China (USD 5.1 billion) in terms of annual sales. Nevertheless, prior studies on the DEA model in the coffee franchising field are scarce [1,11–14]. In particular, there are few studies on the mainstream brand coffee market, which has dramatically increased in recent years. Thus, this study is the first attempt to

compare the metafrontier index values between franchisors and franchisees, two parties to a franchise agreement.

As Lanchimba et al. [6] and Perrigot et al. [29] note, as the efficiency of a franchisee increases, the efficiency of its franchisor should also increase accordingly. However, according to the results of bootstrap DEA in Figure 4, some DMUs belonging to premium coffee chain groups are situated in regions where the efficiency of franchisees and franchisors does not converge. In Figure 4A, under the CRS-based assumption, A(02), A(08), and A(09) are located in the above-average franchisee efficiency and below-average franchisor efficiency zones. Conversely, A(06) and B(11) are in the below-average franchisee efficiency and the above-average franchisor efficiency zones, respectively. In particular, many DMUs in the premium coffee chain group are out of the gray boundaries. There are many more DMUs beyond the gray limits, where the efficiencies of franchisors and franchisees are linearly correlated, as shown in Figure 4B of the VRS-based assumption. Therefore, these coffee chains with a large efficiency discrepancy between franchisors and franchisees need to alter their operational strategies for sustainable growth of the franchising business [29]. Specifically, franchisors require operational alternatives to maximize franchisees' efficiency through more innovative management of franchisees. Moreover, franchisors should implement strategic initiatives to increase their efficiency by adjusting the internal profit structure [10–14,31,32].

*5.2. Practical Implications*

Based on these metafrontier DEA results, we herein offer two practical implications to the coffee shop chain industry. First, premium and mainstream coffee shop groups have heterogeneous technical efficiency frontiers, according to their franchise operational initiatives, service plans, price range, and customers' motivation. Thus, we demonstrated whether there is a difference between efficiencies of premium and mainstream coffee brands in a franchise group by using the Mann–Whitney U test. This may explain the difference in efficiency identified by Park et al. [1] among the Korean coffee chain brands. This result shows that there is a significant difference between premium and mainstream coffee chains from the perspective of franchisors. Under the assumptions of CRS, the efficiency of mainstream coffee brands was higher than that of premium coffee brands. This result addresses that premium and mainstream coffee franchisors require different initiatives tailored to their operational strategies. Premium coffee chains need a store-centric high-end strategy with certified baristas and specialty grade beans, while mainstream coffee chains should develop sophisticated marketing and operational plans that allow customers to more easily access the product via price differentiation and locational accessibility.

Second, primary causes underlying inefficiency differ between franchisors and franchisees. From the perspective of franchisors, the main driver of inefficiency in premium coffee chains is attributed to scale inefficiency, with the bulk of franchisors showing decreasing returns to scale. Therefore, premium coffee franchisors require restructuring and downsizing their scale of operations to achieve scale optimization. Meanwhile, inefficiency of franchisees is mainly due to pure technological inefficiency, which means that coffee shop franchisees failed to deploy service resources efficiently and had poor input utilization. Thus, coffee franchisees with managerial inefficiency need innovative store operation plans that can reduce unnecessary waste of resources and promote customer visits. Consequently, this study scrutinized the primary causes of inefficiency in 9 premium and 19 mainstream coffee chain groups and offered a sophisticated approach for achieving optimal economies of scale to improve operational efficiency.

**6. Conclusions and Future Research**

This study employed the metafrontier DEA model to compare and contrast the operational ME of premium and mainstream coffee chain groups. It also empirically investigated how franchisees and franchisors are interrelated in efficiency. Based on the bootstrap DEA

results, this study identified that the franchisor, the principal of the franchise agreement, has different goals and directions from the franchisee as an agent.

In particular, the popularity of low-price mainstream coffee franchises in Korea has seen a sharp upward trend since 2015. These low-price coffee franchisors increase the number of outlets through the competitive advantage of low coffee prices, thereby enhancing the income from the franchise's initial investment. However, as the number of outlets increased, the franchisors' management capacities for coffee outlets were exhausted, resulting in poor franchisee management. This reduces franchisees' revenue and lowers their operating efficiency. For example, in *XOXO HOTDOG & COFFEE* (B_11), *YOGER-PRESSO* (B_12), and *Cheongja Dabang* (B_13), the efficiency of the franchisor is high, but that of the franchisees is low. These are examples of coffee franchises demonstrating a large discrepancy in efficiency between the franchisor and franchisees in the franchise agreement of the principal–agent relationship.

Meanwhile, DROPTOP (A_02), TWOSOME PLACE (A_08), and HOLLY COFFEE (A_09) in the premium group are coffee franchises with high franchisee efficiency but low franchisor efficiency. These coffee franchises in particular face challenges in that the total sales of franchisors are slight, and the number of outlets is low compared to the projection. In the instance of DROPTOP (A_02), the number of outlets (actual output = 219) is approximately 18.63% of the projected outlets (projection output = 1176). To maintain their intrinsic premium characteristics, these premium coffee franchises seek to increase royalty income by boosting the franchisees' profitability rather than earning back the franchise's initial fee by launching a new coffee outlet. Consequently, these coffee franchises have a relatively low output compared to their input, and it is necessary to improve their operational efficiency by reducing excessive inputs.

While this study provides meaningful insights for both the coffee franchise management theory and practice, it has some limitations. First, this study measured the operational efficiency of coffee franchisors and franchisees but did not analyze the relationship between their operational efficiency and internal operating factors or external environmental variables. From the perspective of franchisors, changes in environmental factors such as the type of franchising contract between franchisors and franchisees (e.g., the method of profit sharing or procurement of goods/supplies) and the revision of franchising-related laws have an enormous impact on franchisors' operational efficiency. In addition, from the customers' point of view, factors such as geographic accessibility of a store, variety of menus, coffee pricing ranges, and store ambiance have a large impact on the operational efficiency of individual coffee franchises. Therefore, future studies should find factors that affect the operational efficiency of coffee franchises and analyze how this effect manifests. Second, this study measured the relative efficiency of franchisors and franchisees using financial data of coffee franchises in 2020. However, during this period, social distancing was at its peak due to the COVID-19 pandemic, and food intake in coffee-shops was restricted. Moreover, external factors such as COVID-19 generated an extraordinary atmosphere imposing numerous constraints on coffee franchise operations. Thus, it is somewhat difficult to generalize the results of the operational efficiency of franchises during these turbulent times. Further studies should consider excluding periods with strong external environmental factors.

**Author Contributions:** All authors worked collectively and significantly contributed to this paper. Conceptualization, S.B.K. and K.C.; methodology, S.B.K. and K.C.; validation, S.B.K. and K.C.; writing, S.B.K. and K.C.; suggestion, S.B.K. and K.C.; data gathering, S.B.K. All authors have read and agreed to the published version of the manuscript.

**Funding:** This research received no external funding.

**Institutional Review Board Statement:** Not Applicable.

**Informed Consent Statement:** Not Applicable.

**Data Availability Statement:** Not Applicable.

**Acknowledgments:** This research was financially supported by Hansung University.

**Conflicts of Interest:** The authors declare no conflict of interest.

## Appendix A  The categorization, DMU codes, and full names of the decision making units (DMUs)

| Group | DMU Code | Coffee Chain Name | Business Start Date |
|---|---|---|---|
| Premium Coffee Franchise group | A(01) | Dal.komm Coffee | 7 May 2012 |
| | A(02) | DROPTOP | 18 Aug 2011 |
| | A(03) | EDIYA COFFEE | 17 Aug 2001 |
| | A(04) | Caffé TIAMO | 1 Apr 2005 |
| | A(05) | CAFÉ-BENE | 28 Jun 2008 |
| | A(06) | Coffeenie | 5 Nov 2009 |
| | A(07) | TOMNTOMS | 18 Dec 2004 |
| | A(08) | TWOSOME PLACE | 30 Sep 2008 |
| | A(09) | HOLLYS COFFEE | 1 Jun 1999 |
| Mainstream Coffee Franchise group | B(01) | THE LITTER | 1 Aug 2015 |
| | B(02) | the Venti | 21 Mar 2014 |
| | B(03) | DUTCH&BEAN | 4 Nov 2014 |
| | B(04) | 10000LAB COFFEE | 14 Oct 2015 |
| | B(05) | MEGA MGC COFFEE | 9 Mar 2016 |
| | B(06) | BULK COFFEE | 15 Jan 2018 |
| | B(07) | Café BOMBOM | 16 Jan 2015 |
| | B(08) | PAIK'S COFFEE | 7 Sep 2009 |
| | B(09) | Selecto Coffee | 20 Nov 2012 |
| | B(10) | Amasvin | 12 Aug 2008 |
| | B(11) | XOXO HOTDOG & COFFEE | 22 Apr 2016 |
| | B(12) | YOGER-PRESSO | 20 Nov 2007 |
| | B(13) | Cheongja Dabang | 14 Sep 2015 |
| | B(14) | Coffee-mama | 8 Sep 2010 |
| | B(15) | COFFEE-BAY | 30 Jun 2011 |
| | B(16) | Coffee-banhada | 30 Mar 2011 |
| | B(17) | COFFEE ONLY | 10 Sep 2016 |
| | B(18) | Compose Coffee | 11 Aug 2014 |
| | B(19) | ToPresso | 1 Oct 2005 |

## Appendix B  Mathematical framework for Metafrontier Efficiency

In the section, we introduce metafrontier model proposed by O'Donnell et al. [12], Battese [13], and Battese and Rao [14], the mathematical framework for assessing the metafrontier DEA is as follows. Let us assume that units use a particular output vector, $y$, can be produced using a given input vector, $x$, in any one of the groups, we consider that $(x, y)$ belongs to the metafrontier, $T^*$. The input and output sets associated with the metafrontier set can be written as follows:

$$P(x) = \{y : (x, y) \in T\} \tag{1}$$

Let $D_o^k(x, y)$ indicate the output-oriented distance function of $k$th group can be given as:

$$D_o^k(x, y) = \inf_\theta \left\{ \theta > 0 : \left( \frac{y}{\theta} \right) \in P^k(x) \right\} \tag{2}$$

To ensure the convexity property [12], the metafrontier is defined as the convex hull of the union of group-specific technologies, denoted by:

$$T^* \supseteq \text{Convex Hull} \left\{ T^1 \cup T^2 \cup \cdots T^k \right\} \tag{3}$$

Let $D^*(x, y)$ denote the output-distance functions defined using the metatechnology, $T^*$. Following the definition of the metatechnology, we can easily establish the following results:

$$D^k(x, y) \geq D^*(x, y), \quad k = 1, 2, \cdots, K \tag{4}$$

An output-oriented measure of technical efficiency of with respect to group $k$ technology for a pair $(x, y)$ is defined as:

$$TE_o^k(x, y) = D_o^k(x, y) \tag{5}$$

The output-orientated technology gap ratio for $k$th group can be defined using the output distance functions from technologies $T^k$ and $T^*$ as [14]:

$$TGR_o^k(x, y) = \frac{D_o^*(x, y)}{D_o^k(x, y)} = \frac{TE_o^*(x, y)}{TE_o^k(x, y)} \tag{6}$$

Equation (6) provides a convenient decomposition of the technical efficiency of a particular input-output combination, relative to that of group $k$:

$$TE_o^*(x, y) = TE_o^k(x, y) \times TGR_o^k(x, y) \tag{7}$$

which shows that technical efficiency measured with reference to the metatechnology can be decomposed into the product of the technical efficiency and the technology gap ratio measured with reference to the group $k$ technology. To estimate the efficiency score, the output-oriented linear program is represented as follows:

$$\left\{ D_o^k(x_m, y_m) \right\}^{-1} = \max \theta$$

$$s.t. \sum_{i=1}^{I} z_i \cdot y_{im} \geq \theta \cdot y_{im}, \quad m = 1, 2, \cdots, M$$

$$\sum_{i=1}^{I} z_i \cdot y_{in} \leq x_{in}, \quad n = 1, 2, \cdots, N \tag{8}$$

$$z_i \geq 0, \quad i = 1, 2, \cdots, I$$

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
