# Peer review of "Bridging the Operational Efficiency Differences between Franchisors and Franchisees: A Metafrontier Approach"

_processes, doi:10.3390/pr10102021_

Round 1

Reviewer 1 Report

The subject of the article is interesting, and it is linked to the objectives of the journal, however, there are some issues that have to be reconsidered.

For better visibility on databases, the authors are asked not to repeat among keywords the words/concepts included in the title of the article.

I suggest that the part called Literature Review of Franchise Efficiency should be enriched with studies from outside of Asia.

A better explanation on how those 28 coffee chain brands (9 premium brands and 19 mainstream brands) were slected as DMUs is required. Why do the authors consider they are representative of the entire sector?

If the Discussion and Conclusion would be splitter into 2 distinctive parts, it could help better understand the research. 

Author Response

Authors’ Response to the Review #1 Comments

Journal: Processes

Manuscript #: processes-1936321

Title of Paper: Bridging the operational efficiency differences between franchisors and franchisees: A meta-frontier approach

Date Sent: 2022-09-10

First, we want to thank the reviewers for their positive and constructive evaluation of our manuscript amid their busy schedules. We have done our best to reflect all the comments by the reviewers in the revision, and we feel that this has improved the manuscript’s quality greatly, for which we are very grateful.

(Comments 1) For better visibility on databases, the authors are asked not to repeat among keywords the words/concepts included in the title of the article.

(Answer) Thank you for your kind comment. I agree with your suggestions. Based on your comments, we changed keywords as follow;

(First draft) metafrontier; coffee shop franchise; franchisor; franchisee; Mann-Whitney U test,

(Revised draft) coffee franchise; luxury and mainstream brand; metafrontier DEA; Mann-Whitney U test

(Comments 2) I suggest that the part called Literature Review of Franchise Efficiency should be enriched with studies from outside of Asia.

(Answer) The coffee franchise sector is one of the major subjects covered in the first draft's literature. Followed by your comments, this revised manuscript extends the scope of the previous literature reviews to the food chain industry (e.g., food restaurant chain, and coffee franchise) by considering the topics of special issues of the "Processes" journal.

(Revised page/lines) Page 3, Line 104-107, 131-139, Table 1 (red letter).

(Comments 3) A better explanation on how those 28 coffee chain brands (9 premium brands and 19 mainstream brands) were selected as DMUs is required. Why do the authors consider they are representative of the entire sector?

(Answer) Actually, there are more coffee franchises in Korea than the DMUs of coffee chains adopted in this study. However, this study used all coffee franchises highly ranked in the Korean coffee chain as DMUs, except for some franchises that do not disclose financial and non-financial information (e.g., Starbucks, PaulBasset, and Caffe-Pascucci). This was due to the difficulty of data accessibility. Followed by your comments, we added some sentences related to the adoption criteria for DMU. 

(Revised page/lines) Page 5, Line 150-153

(Comments 4) If the Discussion and Conclusion would be splitter into 2 distinctive parts, it could help better understand the research. 

(Answer) Followed by your suggestions, this revised manuscript splits ‘5. Discussion and Conclusions (first draft)’ into two section: ‘5. Discussion’ and ‘6. Conclusions & Future Research’. Furthermore, we added practical examples based on the results of metafrontier DEA to the conclusion section to provide some managerial insights.

(Revised page/lines) Page 12-14, Line 432-459.

Reviewer 2 Report

Nothing to comment on. The article is interesting and well-structured. It contains all the main sections and they are well-explained and referenced.

Author Response

Thank you for your favourable comment. 

Reviewer 3 Report

This paper attempts to discuss an interesting question about the relationship between coffee franchisees and franchisors in Korea by using the meta-frontier data envelopment analysis (DEA) and bootstrap DEA. Overall, this is an interesting research subject. However, I have some concerns that should be further addressed.

1. Some sentences in Abstract part is too long to understand, such as “Nonetheless, the question........” in line 2 to 4. Moreover, the results of this paper should be mentioned briefly and clearly in Abstract.

2. Introduction section should be strongly strengthened. The writing of this paper is too rough and not grammatically presented, such as “an increase of ........” in line 6 of the first paragraph. The background of this work should be well-organized. In addition, I suggest that the contributions of this work should be presented in the introduction section.

3. The literature review section is too short, which should be enhanced. The authors should update the latest data and literatures in the introduction and literature review sections, such as doi: 10.1016/j.cie.2020.106951. In addition, literature review should not a simple stack of papers, but a comprehensive analysis. I suggest that the author should organize and summarize relevant literatures.

4. In the results section, I think that the authors should provide sufficient examples or literature comparisons to support your results and findings, which may highlight your contributions.

5. Conclusion section is too weak and it should be should be strongly strengthened. I suggest that the author should provide some managerial insights from the results. Moreover, the practical examples should also be added to the conclusion section.

6. The logic of this paper is not very clear, and the language should be improved significantly. Please check the formula symbols and typos in the paper to ensure that they are correct.

Author Response

Authors’ Response to the Review #3 Comments

Journal: Processes

Manuscript #: processes-1936321

Title of Paper: Bridging the operational efficiency differences between franchisors and franchisees: A meta-frontier approach

Date Sent: 2022-09-10

First, we want to thank the reviewers for their positive and constructive evaluation of our manuscript amid their busy schedules. We have done our best to reflect all the comments by the reviewers in the revision, and we feel that this has improved the manuscript’s quality greatly, for which we are very grateful.

(Comments 1) Some sentences in Abstract part is too long to understand, such as “Nonetheless, the question........” in line 2 to 4. Moreover, the results of this paper should be mentioned briefly and clearly in Abstract.

[ (Answer) Thank you for your kind comment. I agree with your suggestions. Based on your comments, we shortened the length of the sentences and inserted main results of this study in Abstract section.

(Revised page/lines) Page 1, Line 10-14, 15-22.

(Comments 2) Introduction section should be strongly strengthened. The writing of this paper is too rough and not grammatically presented, such as “an increase of ........” in line 6 of the first paragraph. The background of this work should be well-organized. In addition, I suggest that the contributions of this work should be presented in the introduction section.

[ (Answer) Followed by your comments, we modified this sentences (line 30-32) and inserted summarized theoretical and practical contribution in the introduction section. 

(Revised page/lines) Page 3, Line 10-14, 16-22.

(Comments 3) The literature review section is too short, which should be enhanced. The authors should update the latest data and literatures in the introduction and literature review sections, such as doi: 10.1111/itor.13186 and doi: 10.1016/j.cie.2020.106951. In addition, literature review should not a simple stack of papers, but a comprehensive analysis. I suggest that the author should organize and summarize relevant literatures.

[ (Answer) The coffee franchise sector is one of the major subjects covered in the first draft's literature. Followed by your comments, this revised manuscript extends the scope of the previous literature reviews to the food chain industry (e.g., food restaurant chain, and coffee franchise) by considering the topics of special issues of the "Processes" journal. Additionally, we updated the introduction and literature review sections with the latest literatures including the two papers you recommended.

(Revised page/lines) Page 3, Line 104-107, 131-139, Table 1 (red letter).

(Comments 4) In the results section, I think that the authors should provide sufficient examples or literature comparisons to support your results and findings, which may highlight your contributions.

[ (Answer) Thank you for your kind feedback. I agree with your advice. Thus, we added some literature to support our results and contributions.

(Revised page/lines) Page 7, Line 241-243, 249-253, 335-338.

(Comments 5) Conclusion section is too weak and it should be should be strongly strengthened. I suggest that the author should provide some managerial insights from the results. Moreover, the practical examples should also be added to the conclusion section.

[ (Answer) Followed by your suggestions, this revised manuscript splits ‘5. Discussion and Conclusions (first draft)’ into two section: ‘5. Discussion’ and ‘6. Conclusions & Future Research’. Furthermore, we added practical examples based on the results of metafrontier DEA to the conclusion section to provide some managerial insights.

(Revised page/lines) Page 12-14, Line 432-459.

(Comments 6) The logic of this paper is not very clear, and the language should be improved significantly. Please check the formula symbols and typos in the paper to ensure that they are correct.

[ (Answer) This study used professional editing services to improve the readability of readers. And we attached the certification of Editage service.

Round 2

Reviewer 1 Report

The authors succeed in answering my concerns, so the article can be publish. 

Reviewer 3 Report

The authors of the paper have revised according to the revision comments, the quality of the paper has been improved, and it is recommended to accept and publish.